



# Evidence of successful methane mitigation in one of Europe's most important oil production region

Gerrit Kuhlmann[1], Foteini Stavropoulou[2,3], Stefan Schwietzke[3], Daniel Zavala-Araiza[3], Andrew Thorpe[4], Andreas Hueni[5], Lukas Emmenegger[1], Andreea Calcan[6,7], Thomas Röckmann[2], and Dominik Brunner[1]

[1]Empa, Laboratory for Air Pollution / Environmental Technology, Dübendorf, Switzerland
[2]Institute for Marine and Atmospheric Research Utrecht, Utrecht University, Utrecht, The Netherlands
[3]Environmental Defense Fund, Berlin, Germany
[4]Jet Propulsion Laboratory, California Institute of Technology, Pasadena, California, United States
[5]Remote Sensing Laboratories, Department of Geography, University of Zurich, Zurich, Switzerland
[6]INCAS, National Institute for Aerospace Research "Elie Carafoli", Bucharest, Romania
[7]International Methane Emissions Observatory (IMEO), UNEP, Paris, France

**Correspondence:** Gerrit Kuhlmann (gerrit.kuhlmann@empa.ch)

**Abstract.** Reducing methane emissions from the oil and gas production infrastructure is a cost-effective way for limiting global warming. In 2019, a measurement campaign in southern Romania found emission rates from the oil and gas sector substantially higher than the nationally reported emissions with a few high-emitting sources ("super-emitters") contributing disproportionately to total emissions. In 2021, our follow-up airborne remote sensing campaign, covering over 80% of production sites, revealed a marked decrease in super-emitters. The observed change in the number of emitters is consistent with an emission reduction by 20-60% from 2019 to 2021. This reduction is likely due to improvements in production infrastructure following the first campaign in 2019. This is further supported by additional site visits, which showed that many of the leaks identified in 2019 had indeed been mitigated. However, our top-down quantification remains higher than the bottom-up emission reports. Our study highlights the importance of measurement-based emission monitoring of climate change mitigation measures, and illustrates the value of a multi-scale assessment integrating ground-based observations with large-scale airborne mapping to capture both the primary mode of emission sources and the rare, but significant, super-emitters.

## 1 Introduction

In 2015, the United Nations Climate Change Conference (UNFCCC) in Paris agreed to limit global warming well below 2 °C, which requires massive reductions of greenhouse gas (GHG) emissions (UNFCCC, 2015). Methane ($CH_4$) is a major contributor to global warming and an attractive near-term target for climate change mitigation. In 2021, this was acknowledged in the Global Methane Pledge (GMP) (European Commission and United States of America, 2021). In particular, $CH_4$ emission reductions related to oil and gas (O&G) infrastructure are considered "no-regret" solutions, as they have mainly positive effects and can often be realized in a cost-effective way (Hopkins et al., 2016; McKain et al., 2015). In the EU-27, $CH_4$ emissions from O&G production constitute about one third of the total reported emissions caused by the energy sector (UNFCCC, 2023a). The



International Energy Agency (IEA) estimates that Romania's onshore O&G production sector has the highest share of all EU countries, contributing 16% to the total onshore O&G emissions (IEA, 2021), even after reported emissions have allegedly decreased by more than 85% since 1990 (UNFCCC, 2023b), However, reported $CH_4$ emissions are highly uncertain as direct $CH_4$ emission measurements to independently verify these numbers are lacking.

To close this gap, the ROmanian Methane Emissions from Oil and gas (ROMEO) measurement campaign provided the first independent estimates of $CH_4$ emission rates from O&G production in Romania. The campaign took place in October 2019 and covered the southern part of the country with ground, drone, and aircraft in situ measurements (Delre et al., 2022; Korbeń et al., 2022; Stavropoulou et al., 2023; Maazallahi et al., 2024). The national-scale emissions derived from the campaign for oil production sites were already higher than the total reported emissions for the entire O&G sector in Romania. Furthermore, 10% of the sites emitted more than 10 kg/h, accounting for more than 70% of the total emissions (Stavropoulou et al., 2023). To accurately determine the total $CH_4$ emissions, it is therefore critical to detect and quantify a statistically robust number of high-emitting sources ("super-emitters"). However, this is challenging with ground-based surveys due to the large number of production sites and associated infrastructure distributed over a vast and sometimes difficult-to-access area. In 2021, we therefore deployed the airborne AVIRIS-NG imaging spectrometer in southern Romania. AVIRIS-NG provides excellent spatial coverage with a detection limit ($>10$ kg/h) well suited for detecting the emissions of super-emitters (e.g. Thorpe et al., 2013; Frankenberg et al., 2016; Thorpe et al., 2017; Foote et al., 2020; Borchardt et al., 2021).

In this work, we present the results from the AVIRIS-NG measurement campaign in 2021. The AVIRIS-NG measurements are used to detect and quantify the largest $CH_4$ sources in the study area. The results were integrated with the ground-based data from 2019 to more accurately constrain the annual $CH_4$ emissions of the O&G sector in the country.

## 2 Data and Methods

### 2.1 AVIRIS-NG campaign in 2021

The AVIRIS-NG flights were conducted on Thursday, 29[th] and Friday, 30[th] July 2021 over the same region as the ROMEO campaign in 2019 (Fig. 1a). AVIRIS-NG is a push-broom imaging spectrometer sampling 600 pixels in across-flight direction over a 34° field of view (Hamlin et al., 2011). This results in a ground pixel size of about 5 m for an altitude of 6000 m above ground. The spectrometer covers a spectral range from 380 to 2510 nm at 5 nm sampling resolution, which includes two spectral windows with $CH_4$ absorption lines at 1.6 $\mu$m and 2.4 $\mu$m. The signal-to-noise ratio (SNR) in these windows is about 400 for a solar zenith angle of 30° and surface reflectance of 30% (Cusworth et al., 2019).

The flights covered an area of approximately 3000 km$^2$ that includes 66% of known processing facilities and 82% of O&G production sites in the region. This translates to 582 processing facilities, and 2805 oil and 299 gas production sites (Table S3). Weather conditions were mostly cloud-free with low wind speeds of about 0.5 m s$^{-1}$ in the morning (8-10 UTC) of the first day that increased to 3.0 m s$^{-1}$ by noon and remained high on the second day. Some lines were flown twice due to the presences of scattered clouds during first data collection.





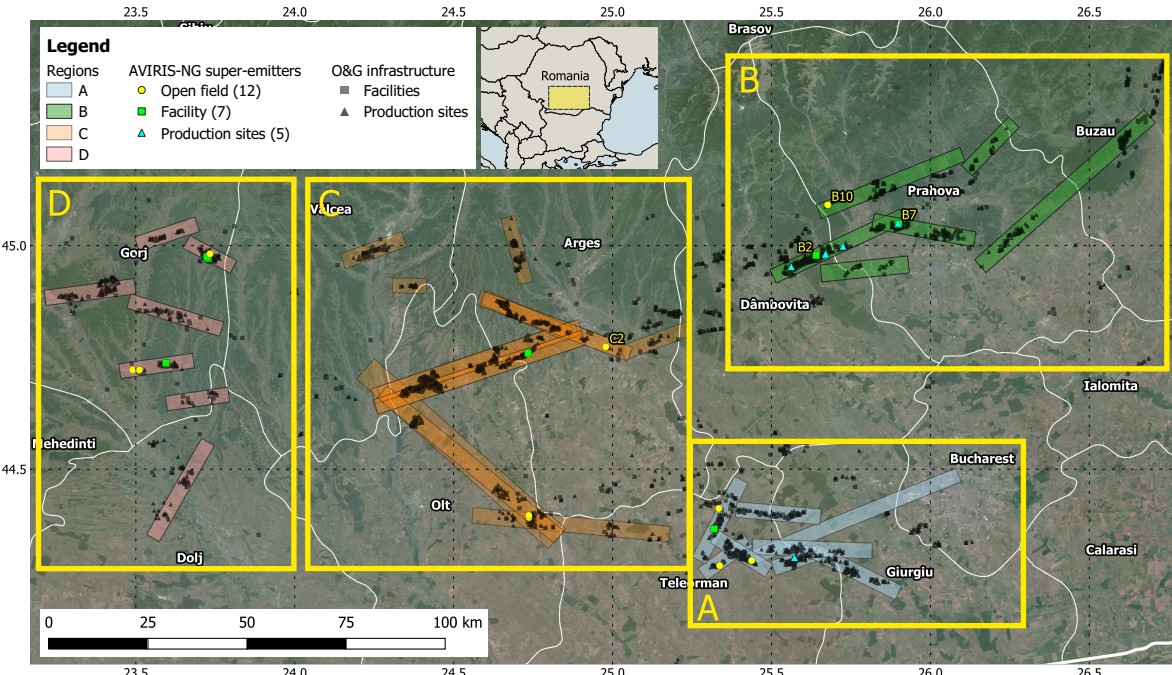

**Figure 1.** Map with AVIRIS-NG flight lines in 2021 for four regions (A-D) with locations of super-emitters and O&G infrastructure. Background map from the Copernicus Land Monitoring Service (GioLand/VHR_2018_WM).

## 2.2 Methane retrieval algorithm

### 2.2.1 Matched filter

$CH_4$ enhancements were retrieved from the AVIRIS-NG radiance cube using the well-established matched filter method following the implementation by Foote et al. (2020). The matched filter $\boldsymbol{q}$ is a linear filter that is applied to each radiance spectrum $\boldsymbol{y}_i$ to obtain a scalar $\alpha_i$, which is the enhancement in the $CH_4$ column density above the background in units of ppm m. For a linear radiance model, we can write the product

$$\boldsymbol{q}^T \cdot \boldsymbol{y}_i = \boldsymbol{q}^T \cdot (\boldsymbol{\mu} + \alpha_i \boldsymbol{t} + \boldsymbol{\varepsilon}) \tag{1}$$

where $\boldsymbol{\mu}$ is the mean radiance spectrum, $\boldsymbol{t}$ is the $CH_4$ target signature and $\boldsymbol{\varepsilon}$ is the remaining clutter in the spectrum including instrument errors and variability of the surface reflectance. The solution for the optimal matched filter maximizes the signal (second term in brackets) and minimizes the clutter (third term). The $CH_4$ column enhancement is then

$$\alpha(\boldsymbol{y}_i) = \frac{(\boldsymbol{y}_i - \hat{\mu})^T \cdot \hat{\mathbf{S}}^{-1} \cdot \boldsymbol{t}}{\boldsymbol{t}^T \cdot \hat{\mathbf{S}}^{-1} \cdot \boldsymbol{t}}, \tag{2}$$

where $\hat{\mu}$ and $\hat{\mathbf{S}}$ are the mean vector and the covariance matrix estimated from the radiance cube.



A suitable target signature $\boldsymbol{t}$ can be obtained by computing the change in the at-sensor radiance spectrum $L_\varepsilon$ due to an enhancement in $CH_4$ absorption using Lambert-Beer's law

$$L_\varepsilon(\lambda) = L_0(\lambda) \exp(-\alpha_\varepsilon s(\lambda)) \tag{3}$$

where $s(\lambda)$ is the unit absorption spectrum in $(ppm\,m)^{-1}$ and $L_\varepsilon(\lambda)$ and $L_0(\lambda)$ are the radiance spectra with and without a $CH_4$ enhancement. For a small absorption, Equation (3) can be approximated by

$$L(\lambda) \approx L_0(\lambda) + L_0(\lambda)\alpha s(\lambda), \tag{4}$$

which, if $L_0(\lambda)$ is approximated by the mean spectrum $\mu(\lambda)$, gives the target signature from Eq.(1) as:

$$t(\lambda) = s(\lambda) \cdot \mu(\lambda). \tag{5}$$

The unit absorption spectrum $s$ can be computed from Eq. (3) as the change in the natural logarithm of at-sensor radiance spectrum $L_\varepsilon$ due to an enhancement of 1 ppm in the surface layer:

$$s(\lambda) = \frac{\ln L_0(\lambda) - \ln L_\varepsilon(\lambda)}{\alpha_\varepsilon}. \tag{6}$$

In this study, we used a simplified but fast forward model that ignores atmospheric scattering to compute the at-sensor radiance for a Lambertian surface

$$L(\lambda) = \mu_0 E_0(\lambda)\frac{\rho}{\pi} \cdot e^{-m\left(\left(1+\frac{x_\varepsilon}{x_t}\right)\tau_t(\lambda)+\tau_{BG}(\lambda)\right)} \tag{7}$$

with the cosine of the solar zenith angle $\mu_0$, solar irradiance spectrum $\boldsymbol{E}_0$ (Coddington et al., 2021), surface reflectance ($\rho = 1.0$) and the geometric air mass factor for a nadir-viewing instrument $m$. The $CH_4$ optical depth in the surface layer (i.e. below 1000 m) is given by $\tau_t(\lambda)$. $\tau_{BG}(\lambda)$ is the optical depth of all other gases including the part of the $CH_4$ column above the surface layer. $x_\varepsilon$ is the $CH_4$ enhancement (in ppmv) above the $CH_4$ background concentration $x_t$.

The optical depth profiles are computed for a mid-latitude summer reference atmosphere using the atmospheric radiative transfer simulator (arts, Version 2.2, (Buehler et al., 2018)) with absorption lines from the HITRAN 2012 database (Rothman et al., 2013). The profiles are scaled to obtain column-dry averaged $CO_2$ and $CH_4$ mixing ratios of 416 ppm and 1.90 ppm for 2021, respectively. Surface elevation and aircraft altitude for computing the geometric air mass factor are available from the AVIRIS-NG dataset.

Figure 2 shows an example of the unit absorption spectrum computed with the above model equation (dashed line) for the two spectral windows (1480 – 1800 nm and 2080 – 2450 nm) used for the $CH_4$ retrieval. In addition, the range of the unit absorption spectra computed with the detailed libRadtran radiative transfer model (Emde et al., 2016) is shown for a scattering atmosphere with varying aerosol optical depths (AOD) and surface reflectance (RHO).

**Application to AVIRIS-NG radiance cubes**

We apply the matched filter simultaneously to both spectral windows shown in Fig. 2 to minimize the noise in the retrieved $CH_4$ field. The matched filter needs to be applied to each across-track line of the AVIRIS-NG independently to avoid stripes





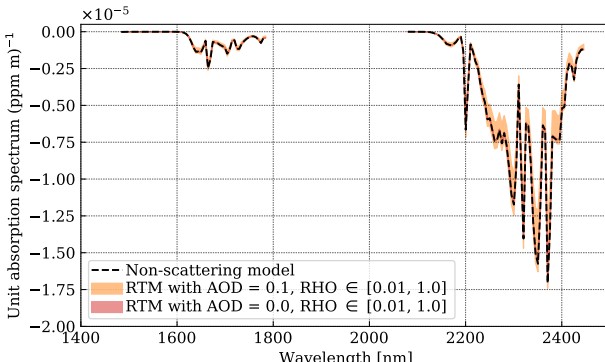

**Figure 2.** Unit absorption spectrum computed with the non-scattering model used in this study compared to libRadtran radiative transfer simulations using different surface reflectance (RHO) and aerosol optical depths (AOD) in two spectral windows.

in the retrieved methane fields caused by small radiometric and spectral calibration differences between these lines, which is

95  problem common to pushbroom imagers. For short lines, we combined across-track positions to have at least 7500 spectra for an unbiased estimate of mean and covariance matrix.

The assumption that $L_0$ can be approximated by the mean spectrum $\boldsymbol{\mu}$ results in a bias in enhancement $\alpha$ that needs be corrected using a scaling factor. Foote et al. (2020) computed the correction factor as the deviation of the radiance spectrum from the mean spectrum:

$$100 \quad \alpha(\boldsymbol{y_i}) = \frac{1}{R_i} \frac{(\boldsymbol{y_i} - \hat{\mu})^T \cdot \hat{\mathbf{S}}^{-1} \cdot \boldsymbol{t}}{\boldsymbol{t}^T \cdot \hat{\mathbf{S}}^{-1} \cdot \boldsymbol{t}} \quad \text{with} \quad R_i = \frac{\boldsymbol{y_i}^T \cdot \hat{\mu}}{\hat{\mu}^T \cdot \hat{\mu}}. \tag{8}$$

Since we apply the matched filter to two spectral windows that can have different surface reflectance, we extend their approach by computing a correction factor for each spectral window separately, which removes biases if surface reflectance strongly differ in the two bands based on tests with synthetic spectra. While the correction factor reduces biases in the $CH_4$ enhancements, it increases retrieval noise for dark surfaces with low signal-to-noise ratio (e.g., forests and water). Note that the second effect

105  can cause the $CH_4$ retrieval to appear to depend on surface reflectance.

### 2.3 Emission quantification

$CH_4$ emissions plumes were identified by visual inspection of the retrieved $CH_4$ maps. Local $CH_4$ enhancements were classified as plume if they were (a) a cluster of pixels with plume-like shape, (b) had significant enhancement above the (local) background variability and (c) did not correlate spatially with surface reflectance or surface features. Once a source had been

110  identified, a plume detection algorithm was used for plume segmentation (Kuhlmann et al., 2019, 2021). The algorithm uses a threshold to identify pixels where the local mean of the signals is significantly enhanced above the background. The threshold was manually adjusted for each plume to fully cover the visible plume and to avoid false positives. A center line was fitted



through the ridge of the detected plume to provide natural coordinates of along- and across-plume direction. The algorithms are part of the open-source Python library for data-driven emission quantification (ddeq; Kuhlmann et al. (2024)).

The CH$_4$ emissions were determined from the detected plume using the integrated mass enhancement (IME, Frankenberg et al. (2016); Varon et al. (2018)) method implemented in the *ddeq* library (Kuhlmann et al., 2024). The IME method is a mass-balance approach derived from a Gaussian plume model that computes the emission rate $Q$ from wind speed $U$, length $L$, the integrated mass enhancement $M$ in the emission plume $\mathcal{P}$ under the assumption of steady-state conditions :

$$Q = \frac{U}{L} \cdot M = \frac{U}{L} \cdot \sum_{i \in \mathcal{P}} A_i \cdot (V_i - V_{i,BG}) \tag{9}$$

where $A_i$ is the pixels size and $V_i$ and $V_{i,BG}$ are enhancement and background CH$_4$ columns of the $i^{th}$ pixel. $L$ is the length of the plume, which is obtained from the arc length of the center line from the source to the last detectable pixel. $U$ is the effective wind speed computed by weighting the wind profile with the CH$_4$ concentration profile (Kuhlmann et al., 2024).

Since many sources identified in this study are located a small vent stacks, the wind speeds were taken from the 10-m wind of the ECMWF operational analysis product at ∼10 km resolution. Its uncertainty was estimated from the ensemble spread of the ERA-5 reanalysis product (∼50 km resolution, (Hersbach et al., 2018)).

While the matched filter retrieves CH$_4$ enhancements above the background in the AVIRIS-NG lines, locally the CH$_4$ background can deviate from zero, for example, due to diffuse CH$_4$ emissions in the area or small systematic errors in the retrieval. Therefore, the local background field $V_{i,BG}$ was computed from the pixels surrounding the detected plume using a normalized convolution with a Gaussian kernel ($\sigma = 3$ pixels), where we masked the convex hull of the detected plume dilated by a disk kernel with a radius of 3 pixels. The dilation was used to avoid overestimating the background by including pixels from the emission plume that where below the detection limit.

The integration area $\mathcal{P}$ contains all pixels of the plume from the source location up to plume length $L$. We computed the integration area as the convex hull of the detected pixel mask dilated by a disk kernel (radius: 3 pixels), selecting an integration area larger than the detected plume. This avoids overestimating the integrated mass $M$ when excluding pixels below the detection limit due to random noise inside the detectable plume. Furthermore, it includes CH$_4$ mass below the detection limit at the plume edges to avoid underestimating the mass $M$. In along-plume direction, the integration area was limited to pixels with an along-plume coordinate between 0 and $L$ to not include pixels upstream of the source and further downstream than $L$.

## 2.4 Estimation of uncertainty

Uncertainties in the estimated emissions are caused by random and systematic errors in the retrieved CH$_4$ enhancements as well as errors in the emission quantification method resulting from errors in the background field, wind speed and plume length as well as methodological limitations of mass-balance approaches such as the assumption of steady-state conditions.

### 2.4.1 Uncertainties in matched filter

The random uncertainty of CH$_4$ enhancements was computed from the AVIRIS-NG SNR by propagation of uncertainty in the matched filter. A SNR about 400 results in a radiance uncertainty of about $12.5 \, \mu$W m$^{-2}$ nm$^{-1}$ sr$^{-1}$. The CH$_4$ uncertainty



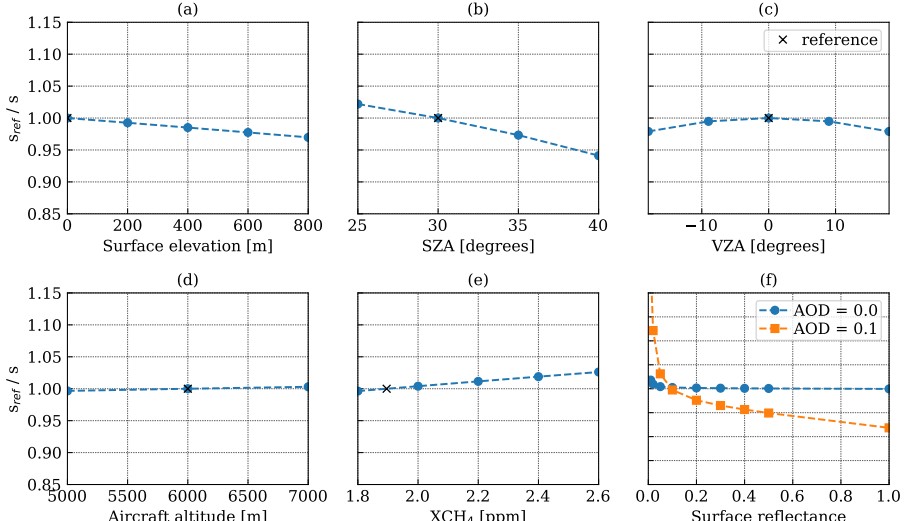

**Figure 3.** Sensitivity of the unit absorption spectrum $s$ to variation of the input parameters computed by varying the parameters of the forward model (Eq. 7). The sensitivity is compared to a reference using surface elevation of $0.0\,\mathrm{m}$, SZA of $30°$, VZA of $0°$, no atmospheric scattering, XCH$_4$ of $1.90\,\mathrm{ppm}$ and aircraft altitude of $6000\,\mathrm{m}$. To test the impact of surface reflectance, $s$ was simulated with libRadtran radiative transfer model using a Rayleigh atmosphere (AOD = 0.0) and aerosol scenario (AOD = 0.1). A value larger/smaller than one would result in over/underestimated CH$_4$ enhancements.

depends on radiance levels due to the scaling of the albedo correction resulting in larger values over dark surfaces and lower values over bright surfaces. The median uncertainty is about $450\,\mathrm{ppmv\,m}$ in good agreement with the standard deviation of the retrieved CH$_4$ field.

     Systematic uncertainty of the CH$_4$ enhancements are caused by simplifications used in the matched filter. We conducted detailed sensitivity tests using synthetic spectra to quantify these errors. The unit absorption spectrum $s$ was computed by

Eq. (7) using mean values for each line, while surface elevation, solar and viewing zenith angles, aircraft altitude and XCH$_4$ background can vary during data acquisition. Figure 3a-e shows that the variability of these parameters during data acquisition does not have a strong impact ($<5\%$) on the unit absorption spectrum and consequently the CH$_4$ enhancements.

     For a non-scattering atmosphere, surface reflectance does not impact the unit absorption spectrum. To analyze the impact of scattering, synthetic spectra were simulated with the libRadtran model (Emde et al., 2016) for varying surface reflectance and

two scattering scenarios. The first scenario only includes molecular scattering setting aerosol optical depth (AOD) to zero. The second scenario uses libRadtran's default aerosol scenario with rural-type aerosols below $2\,\mathrm{km}$ and background aerosols above $2\,\mathrm{km}$ under spring-summer conditions and a visibility of $50\,\mathrm{km}$. AODs were set to $0.10$ in the two spectral windows (1.6 and $2.4\,\mu\mathrm{m}$) to simulate the high aerosol load (up to $0.15$ at $1.6\,\mu\mathrm{m}$ at the AERONET station in Bucharest) due to the presence of Saharan dust in the atmosphere (well-mixed below 4-5 km based on CAMS forecast). Figure 3f shows no significant impact

of molecular scattering alone, while an AOD of $0.10$ results in a ratio varying between $1.10$ to $0.95$ for a surface reflectance





ranging from 0.02 to 1.0. Since over darker surfaces (<0.02), $CH_4$ random uncertainty is too high for identifying plumes, systematic uncertainties due to aerosols are expected to be smaller than ±5% for the detected plumes during the campaign.

Further potential sources of systematic errors are the computation of mean vector $\hat{\mu}$ and covariance matrix $\hat{\mathbf{S}}$. Since $\hat{\mu}$ and $\hat{\mathbf{S}}$ are computed from a single radiance cube acquired along a flight line, it is necessary to have a sufficiently large number of
spectra for unbiased estimates. Since the matched filter is applied for across-track position, the number can be small for short lines. Sensitivity tests with synthetic spectra showed that 7500 spectra are sufficient (i.e. <2% bias) for an unbiased estimate of 100 pixels with $CH_4$ enhancements of 5000 ppmv m.

The matched filter assumes that the unit absorption depends linearly on the $CH_4$ enhancement, which is not the case for optical thick absorbers. The linearization results in a systematic error in the estimated $CH_4$ enhancements that depends on surface
reflectance and the true enhancement. The bias is generally small for enhancements measured during the campaign (<5%), but can get large for high surface reflectance (>0.50) and strong enhancements (>7000 ppm m), leading to an underestimation of $CH_4$ enhancements by 10-20%. Other trace gases ($CO_2$ and $H_2O$) were found to have no significant impact on the unit absorption spectrum.

Other sources of systematic errors in $CH_4$ retrievals are correlations between the unit absorption spectrum and the wavelength-
dependent surface reflectance. These errors are non-negligible due to the relative low spectral resolution of the AVIRIS-NG instrument but are difficult to quantify (e.g. Ayasse et al., 2018). To avoid impact of surface features on the result, visual inspection of the plumes included ensuring that plumes are not correlated with surface features.

Overall, we estimated that the systematic uncertainty is less than 5% for $CH_4$ fields near identified plumes, while the single sounding precision (random uncertainty) is about 500 ppm m, i.e. about 10% for a median-sized plume with 20 pixels and
enhancement of 1000 ppm m.

### 2.4.2 Uncertainties in the IME approach

Uncertainties in the emission quantification method result from uncertainties in the $CH_4$ retrieval, the background field, wind speed and plume length as well as methodological limitations of mass-balance approaches such as the assumption of steady-state conditions and the assumption that the plume is within the convex hull of the detected pixels. The uncertainty of the plume
length was set to 10% and to at least half a pixel size (i.e. about 5 m).

The uncertainty of the integrated mass $M$ was computed from the random uncertainty of the $CH_4$ retrieval. For the plume identified in the campaign, the mean uncertainty is 11% ranging from 3% to 32% depending on plume size. The mass $M$ is sensitive to the size of dilation kernel used for computing the $CH_4$ background and for defining the integration area $\mathcal{P}$. To quantify the impact, we varied the size of the kernel from 1 to 5 for all detected plumes. As a result, IMEs varied between
2% for large plumes and 49% for small plumes. To account for this uncertainty in the estimated background, for the impact of surrounding pixels on the IME, and for the systematic uncertainty of the $CH_4$ enhancements in the error budget, we increased the uncertainty of the IME computed from the random uncertainty by $\sqrt{2}$ and set the minimum IME uncertainty to 10%. The mean and scatter of the local background is 100±111 ppm m. If the emissions were computed without subtracting the background, estimates would be 10±15% larger.





The uncertainty in the effective wind speed $U$ arises from uncertainties in the ECMWF analysis and the effective plume height. The uncertainty of the 10-m wind speed was estimated from the ensemble spread in the ERA-5 reanalysis. In addition, we add 15% uncertainty for the height-dependency of the plume. To calculate this number, we assumed a logarithmic wind profile for a neutrally stratified surface layer ($\bar{u}(z) = \frac{u_*}{\kappa} \ln(z/z_0)$) with von Karman constant ($\kappa$=0.40), surface roughness ($z_0$=0.10 m) and friction wind speed $u_*$ chosen to match the 10-m wind speed (Jacobson, 2005). For this profile, the effective wind speed for a plume at 5 and 20 m above the ground would 15% smaller or higher, respectively, than the 10-m wind speed.

## 2.5 Additional site visits

To further investigate the sources identified by AVIRIS-NG, a team with an Optical Gas Imaging (OGI) camera carried out follow-up visits to each identified location in November 2022. The team searched the area where the plume was detected to identify the origin of the emissions. A protocol with photos and videos was provided for each site.

In 2023, additional site visits were conducted visiting the largest leaks identified in 2019. The primary goal of these sites visits was to check if the leaks, mostly open ends, were fixed as stated by the operator.

## 2.6 Top-down estimates of total emissions from oil and gas in southern Romania

### 2.6.1 Merging of datasets

To improve the top-down estimates of the total emissions from oil and gas in the southern Romania, we merge the datasets from the ground- and drone-based campaign in 2019 with the airborne AVIRIS-NG campaign in 2021. AVIRIS-NG only observed super-emitters above its detection limits. To calculate the total emissions within the AVIRIS-NG flight lines, we used the following formula:

$$E_l = (1 - z) \cdot N_l \cdot f_{\leq DL}^{emis} \cdot \mathrm{EF} + E_{\mathrm{ANG}}, \tag{10}$$

where $z$ is the zero mode, i.e. the fraction of sites with zero emissions, $N_l$ is the number of sites in the flight lines, $f_{\leq DL}^{emis}$ is the fraction of emissions below the AVIRIS-NG detection limit, EF is the emission factor, i.e. the arithmetic mean of the emissions (in kg/h) of all sites, and $E_{\mathrm{ANG}}$ are the emissions retrieved from the AVIRIS-NG observations. A similar approach was used by Sherwin et al. (2024).

The number of sites in the region was provided by the operators (Table S3). The zero mode for oil production sites was determined as 0.25±0.10 from ground-based measurements in 2019. The uncertainty of the zero mode was estimated from the sensitivity tests conducted by Stavropoulou et al. (2023). The emission factor $EF$ was computed from the emission distribution using different scenarios. The emission fraction $f_{\leq DL}^{emis}$ was computed from the AVIRIS-NG detection limits and the emission distributions.

Monte Carlo simulations were used to compute the 95% confidence intervals (CI) considering the uncertainty of the AVIRIS-NG estimates, the zero mode, the wind speed, and the emission factors of gas production sites and processing facilities. The uncertainty in the emission distribution was accounted for by the four scenarios.





### 2.6.2 AVIRIS-NG detection limits

The AVIRIS-NG detection limit for $CH_4$ plumes depends on pixel size, wind speed and uncertainty in the retrieved $CH_4$ maps. In our study, $CH_4$ emission plumes were detected by visual inspection. The smallest plume identified were about 10 m wide and 30-50 m long for a pixel size of about 5 m. These small plumes were generally compact and did not show visible dispersion
in across-plume direction (e.g., Figure S4). To model such plumes, we simulated plumes using a Gaussian plume model with fixed standard width of 5 m and a random uncertainty of 500 ppmv m.

Figure S1 shows these plumes for varying source strength and wind speed. We compute the noise-free peak signal-to-noise ratio (PSNR) for each plume. Comparing the synthetic plumes with the plumes identified in the AVIRIS-NG lines, we estimate that a PSNR of about 2.0 is a good threshold for the detection limit. For the non-dispersive Gaussian plume model given above,
the detection limit can be computed as a function of wind speed from

$$c_{\text{peak}} = \frac{Q_{\text{DL}}}{\sqrt{2\pi}\sigma_y U} = 2\sigma_{\text{CH}_4}, \tag{11}$$

which gives

$$Q_{\text{DL}} = 2\sqrt{2\pi}\sigma_y\sigma_{\text{CH}_4}U \approx 30.6\,\frac{\text{kg/h}}{\text{m/s}}\cdot U \tag{12}$$

The uncertainty of the detection limit was computed from the variability of the wind speed in the study area during data
acquisition, which is about 35% (Tab. S2).

### 2.6.3 Scenarios for changes in the emission distribution for oil production sites

For the campaign in 2019, Stavropoulou et al. (2023) determined emission distributions for oil production sites using probability density functions that follow log-normal distributions: a mean distribution and two extremes corresponding to a lower and upper limit of the estimated emissions (95% CI). The distributions were fitted as normal distribution with mean $\mu$ and standard
width $\sigma$ by taking the natural logarithm of the $CH_4$ emissions. The emission factor is calculated as $EF = \exp(\mu + 0.5\sigma^2)$ (Stavropoulou et al., 2023).

Since the number of super-emitters identified with AVIRIS-NG was lower in 2021 than expected from these distributions (see results), we considered four scenarios of how the distribution may have changed between 2019 and 2021: Scenario 1 uses that the mean distribution from 2019 and Scenario 2 uses the distribution from 2019 that corresponds to the lower limit of
estimated emissions. Scenario 3 and 4 reduces the standard width $\sigma$ and mean $\mu$, respectively, of the 2019 distribution such that the number of expected emitters matches the six emitters found with AVIRIS-NG. The emission distributions for the four scenarios are shown Figure 4.

### 2.6.4 Emissions from gas production sites and processing facilities

Stavropoulou et al. (2023) only provided the emission distribution and the emission factors for oil production sites, but did not
provide an estimate for gas production sites and processing facilities, as the sample size was smaller. To estimate the emissions





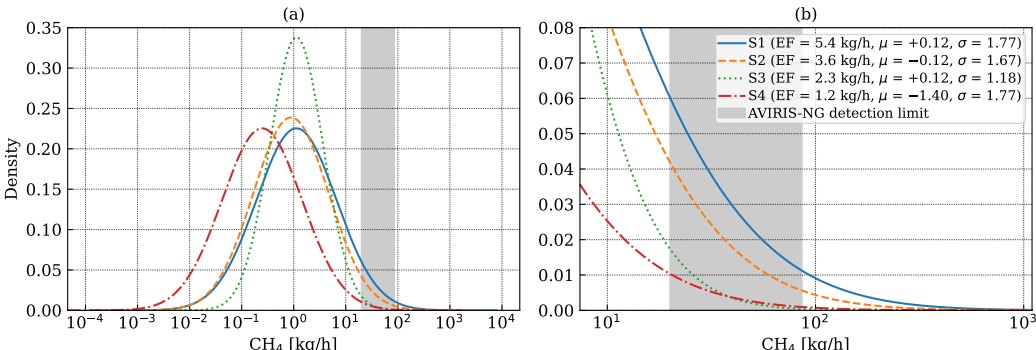

**Figure 4.** The emission distribution of oil production sites for the four scenarios. The AVIRIS-NG detection limit is shown in gray based on varying wind speed during the campaign. Panel (b) presented a zoomed-in view for the long tail of the distribution.

from gas production sites and processing facilities, we assume that the emission distributions for the four scenarios are also applicable for other source categories. In practice, this means that the fraction of emissions below the detection limit $f_{\leq DL}^{emis}$ is the same for all source categories. The emission factors for gas production sites and processing facilities were calculated as arithmetic mean of emission measured during 2019 campaign. These measurements are taken from Table S16 in Stavropoulou
et al. (2023) excluding the estimates using the BDL method.

The emission factors are 11.2±4.0 kg/h for 31 gas production sites and 13.0±3.0 for 60 processing facilities. The 1$\sigma$ uncertainty was estimated using bootstrapping. These emission factors were used for Scenario 1, while for the other scenarios, the emission factors were scaled considering the change of the emission factor for oil production sites for the four scenarios (i.e., 5.4, 3.6, 2.3 and 1.3 kg/h).

**2.6.5    Accounting for varying detection limits by region and day**

Since $f_{\leq DL}^{emis}$ depends on the AVIRIS-NG detection limit, which depends on wind speed, and wind speed varied between regions (A-D in Fig. 1) and days, we computed $f_{\leq DL}^{emis}$ by region and day separately. We thus obtained emission estimates for each region separately with two estimates for region C as measurements were conducted on both days. The wind speed and its variability was computed as mean and standard deviation of the 10-m wind speeds from the ECMWF analysis in the region
during measurement time (Tab. S6). Total emissions are computed by adding up the estimates per region.

**2.6.6    Annual emissions from top-down and bottom-up**

To compute annual emissions (in kt), we assume that the observed emissions are constant in time or at least a representative sample of emissions throughout the year. To compute the total emissions in the study area $E_b$, we scale the emissions inside





the flight lines with the total number of sites in region $N_b$:

$$E_b = \frac{N_b}{N_l} \cdot E_l. \tag{13}$$

## 2.7 Bottom-up estimates of total emissions from oil and gas in entire Romania

Bottom-up reports from oil and gas from energy production in entire Romania were taken from the UNFCCC submission (Table 1.B.2)(UNFCCC, 2023b). We assumed emission categories in Table 1.B.2.a with a description "oil produced" to represent emissions from oil production sites and categories in Table 1.B.2.b with a description "gas produced" to represent emissions from gas production sites. Emissions from venting and flaring correspond to Table 1.B.2.c.

The IEA's estimates for Romanian emissions in 2021 are the sum of categories including 'Onshore Oil', 'Onshore Gas' and 'Gas pipelines and LNG facilities' (IEA, 2021). We group fugitive emissions from onshore oil and gas to oil and gas production sites, respectively. All other emissions are included in "venting and flaring".

## 3 Results

### 3.1 Methane super-emitters identified with AVIRIS-NG

In total, we identified 35 emission plumes at 25 locations, with some sources being observed up to three times due to overlapping flight lines. Maps and photos for all sources are provided in the supplement. The sources were located at oil production sites (6 sources), processing facilities (7) and, unexpectedly, in the open field (12), i.e. not directly linked to O&G infrastructure visible in aerial images. Closer inspection of the sites during the site visits and from images using Google Street View and GoogleEarth suggests that at least six of the open-field sources are linked to O&G processing facilities in the proximity.

Figure 1b and Table S1 show the estimated emission rates for all 25 sources. The source strengths ranged from 16 to 501 kg/h with a median of 80 kg/h (Fig. 1b). The total measured emissions were 2975 kg/h with 16% from oil production sites, 30% from the processing facilities, and 54% from sources located in the open field. The OGI team successfully identified emissions from 14 sources (5 at oil production sites, 6 at processing facilities, and 3 in the open field linked to facilities), indicating that these sites continued to emit over a period of at least one more year. The mean uncertainty of the estimated emission rate is 34% ranging from 22 to 52% .

Figure 6 shows four examples for sources identified in the campaign. For each source, we show an aerial image, the AVIRIS-NG $CH_4$ map with inferred emissions, and a site photograph. Figure 6a shows a $CH_4$ plume at an oil production site in the north-eastern part of the study area. The plume shape (gray dots) is consistent with the westerly wind from the ECMWF analysis product indicated by the arrow. The emission rate was $81\pm22$ kg/h. The plume was also observed one hour earlier in an overlapping flight line with a consistent emission rate ($69\pm18$ kg/h). The site was visited by the OGI team on 22 November 2022. They confirmed the presence of a large $CH_4$ leak originating from an open-ended line as indicated in the photo. An emission plume at an oil processing facility is shown in Figure 6b. The estimated source strength was $90\pm20$ kg/h. The OGI team identified emissions from a vent on the roof of the building on 22 November 2022. Source B10 is located about 100 m



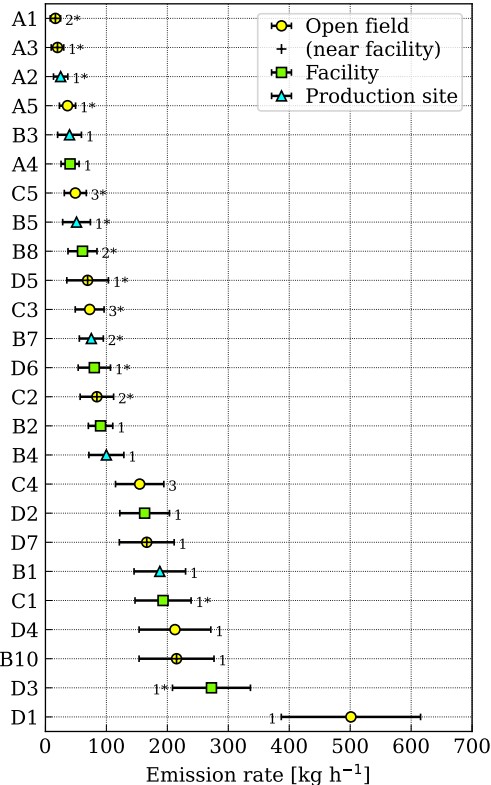

**Figure 5.** Methane super-emitter identified during the airborne campaign with emission rates of the sources with uncertainty ($1\sigma$) and the number of detections. The asterisk marks sources that were also detected by follow-up site visits in 2022.

north of an oil processing facility in the open field (Fig. 6c). The emissions were quantified at $215\pm61$ kg/h. Investigation by the OGI team on November 22, 2021 showed strong emissions originating from a vent stack installed in the field.

    Finally, Source C2 is also located in the open field about 50 m west of a processing facility. The plume was detected both on July 29 and 30, 2021. The site was visited on November 24, 2022, but no source was found. We used the history of Google Street View and Google Earth aerial images to reconstruct the evolution of this facility. On images taken between July 2012

and May 2021, a vent stack in the open field can be seen, which was no longer present in July 2022. Aerial images from February 2021 show heavy construction work on the site, including the installation of a gas flare, which was completed by November 2021. It is likely that the vent stack was dismantled during the same months, but after the AVIRIS-NG flight in July. Consequently, the source could not be located during the site visit in November 2022.




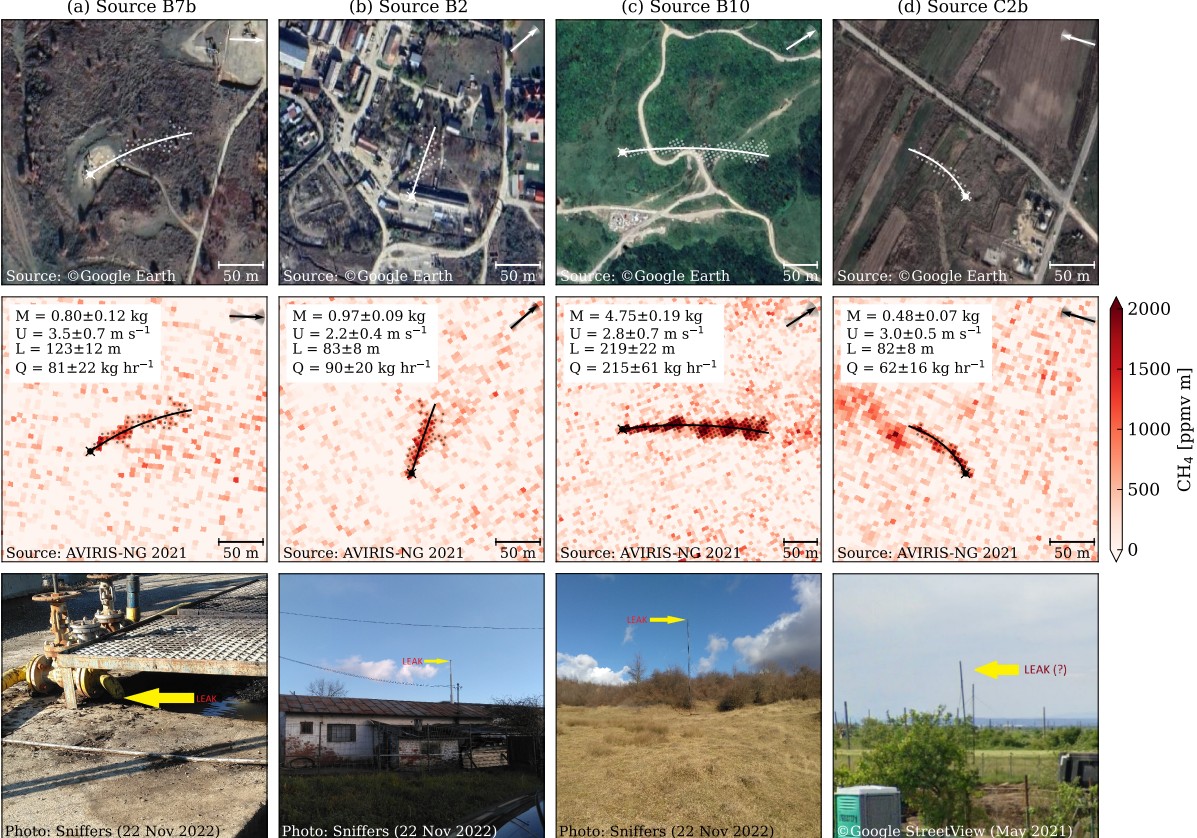

**Figure 6.** Four examples of CH$_4$ plumes at (a) an oil production site, (b) a processing facility and (c and d) from vent stacks in the open field observed in 2021. The panels from top to bottom show Google Earth imagery, AVIRIS-NG CH$_4$ maps with estimated emissions $Q$, and site photos with a yellow arrow indicating the emission point. $M$, $U$ and $L$ are integrated mass, wind speed and plume length. All uncertainties are $1\sigma$. The maps show the detected plumes as dots and the plume center curve as line. The arrow in the top-right corner shows the wind direction from IFS analysis including the spread ($2\sigma$) from the ERA-5 reanalysis as shaded area.

## 3.2 Number of super-emitters in 2019 and 2021

AVIRIS-NG can only measure emissions above its detection limit, which varied between 15 and 107 kg/h during data acquisition, due to varying wind speeds on the campaign days (Eq. 12). Table S6 compares the number of super-emitters expected at oil production sites in each region with the actual number of AVIRIS-NG detections, considering the spatial coverage of the flight campaign. For the mean distribution obtained for 2019 (Scenario 1), we would expect to find between 30 and 190 (95% CI) oil production sites with emissions above the AVIRIS-NG detection limits. However, AVIRIS-NG only detected emissions 320 at 6 oil production sites, which is also less than the 15 to 142 emitters expected from the distribution corresponding to the lower limit of estimated emissions in 2019 (Scenario 2). The number of high-emitting oil production sites was thus significantly lower



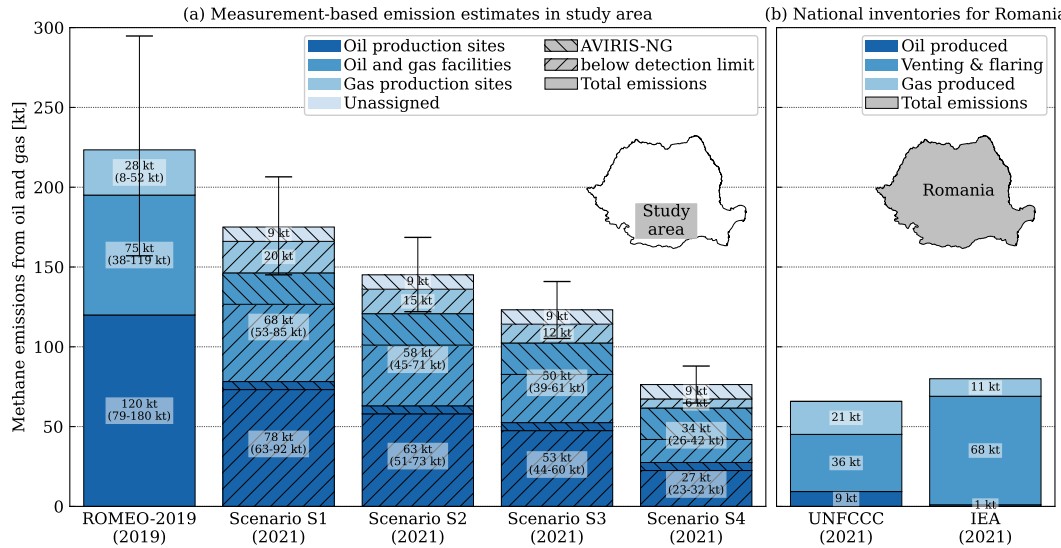

**Figure 7.** Comparison of top-down CH$_4$ emission estimates from O&G infrastructure in the study region with emissions reported to UNFCCC (Table 1.B.2 originating from oil and gas production (UNFCCC, 2023b) and IEA estimates for oil and gas production including pipelines and LNG facilities in entire Romania (IEA, 2021). Top-down estimates are provided for 2019 and, for the four scenarios described in the text, for 2021. Error bars show the 95% confidence interval. "Unassigned" sources are open-field sources that have not been assigned to a processing facility.

in 2021, which suggests that the emission distribution changed between 2019 and 2021. For Scenarios 3 and 4, we adjusted the emission distribution to match the expected number of super-emitters with the detection in 2021.

### 3.3 Total emission in 2019 and 2021

Figure 7a shows our estimates of total emission of CH$_4$ from oil and gas in 2019 and 2021 in southern Romania. Detailed numbers with intermediate steps are shown in the supplement.

For 2019, Stavropoulou et al. (2023) estimated emissions of 120 kt (79-180 kt) from oil production sites. In addition, we estimated 28 kt (8-52 kt) from gas production sites and 75 kt (38-119 kt) from processing facilities in 2019. In 2019, the total emissions from O&G infrastructure were therefore 224 kt (157-295 kt for 95% CI). This estimate from ground and drone-based

measurements agrees well with the independent estimate from airborne in-situ measurements, which is about 227±87 kt for 2019 (Maazallahi et al., 2024).

We estimate that emissions in the study area were 20 to 60% lower in 2021 compared to 2019 with estimates of 175 kt (146-205 kt) in Scenario 1 and 76 kt (66-87 kt) in Scenario 4. Depending on the scenario, super-emitters from AVIRIS-NG account for only 5-14% of total emissions at O&G production sites, but are a major contributor at processing facilities (28-56%).

Consequently, emission reductions are greater at O&G production sites (35-75%) than at processing facilities (10-50%).





To quantify the 95% CIs associated with the expected number of super-emitters and the measurement-based emission estimates, we conducted extensive Monte Carlo simulations, accounting for the uncertainty in the AVIRIS-NG measurements, the AVIRIS-NG detection limits, the emission factors, and the fraction of non-emitting sites. The results of this analysis show that the observed differences between 2019 and 2021 are greater than what can be attributed to the measurement and methodology

uncertainties (95% CI), leading us to conclude that the emission reductions from 2019 to 2021 were significant.

Despite these reductions, the actual methane emissions from oil and gas in the study area in 2021 still exceed the reported emissions for the *entirety* of Romania (Figure 7b). The national inventory report of Romania states that emissions from O&G production were 71 kt in 2019 (not shown) and 66 kt in 2021 split between oil production (14%), gas production (32%), and venting and flaring (54% in 2021) (Table 1.B.2 in UNFCCC Submission 2023 v2). Figure 7b also shows emissions estimated

by IEA, which include onshore oil and gas, pipelines, and LNG facilities. They are 14 kt higher than the emissions reported to the UNFCCC, but still at the lower end of the emissions determined from the measurement campaigns in the study area alone.

## 4 Discussions

The quantification of $CH_4$ emissions from oil and gas is challenging for both bottom-up and top-down approaches due to the limited amount of information available.

The quantification of emissions using the IME method can result in systematic uncertainties from the assumptions made in the implementation. The two largest uncertainties are the effective wind speed and the background field. The effective plume height will generally be higher than the source location due to plume rise and vertical mixing. In this study, we assume a plume height between 5 and 20 m, because 11 out of 25 super-emitters were elevated, mainly, at vent stacks, which are about 5 m high. The remaining source locations closer to the ground (5) or unknown (9). To estimate the background, we decided to subtract the

background field to obtain the local enhancement. The motivation here is that the matched filter retrieves the $CH_4$ enhancement above the AVIRIS-NG line. It is possible that near potential sources, local background concentrations are increased. However, this diffuse emissions cannot be accounted for using the IME method. If we do not subtract the local background, AVIRIS-NG emissions would be 10±15% higher, which within the uncertainty budget assigned to AVIRIS-NG in this study.

The strongest evidence for a change in emissions from 2019 and 2021 is the lower number of super-emitters detected

by AVIRIS-NG at oil production sites. The number depends on the AVIRIS-NG detection limit. Figure S2 shows the sources identified with AVIRIS-NG as a function of wind speed together with the detection limit and its uncertainty from the variability of wind speed. The emission rates are mostly at or above the detection limit. An exception are sources that were already identified in another flight line at lower wind speed (i.e. B7b, C2b and C5c). We therefore conclude that the detection limit is robust estimate. Another reason for the lower number of super-emitters might be seasonal variability in emissions, because

the campaigns in 2019 and 2021 were conducted in July and October, respectively. Data to determine the influence of seasonal effects are unfortunately not available. However, Varon et al. (2023) estimate week-to-week variability for $CH_4$ emissions in the Permian Basin to be about 25%, which, if we assume similar variability in Romania, would be too small to explain the difference between 2019 and 2021 in Romania.





The most likely explanation for the reductions from 2019 to 2021 is that operators implemented improvements in the pro-
duction infrastructure after the 2019 campaign. The operators were informed about the locations of the highest emitting sites
and the likely origin of the emissions (mainly open-ended lines) already in 2020, well before the flights in 2021. During the
two-year gap between the campaigns, operators therefore had the opportunity to implement emission mitigation measures.
In fact, the operators have communicated to us that the infrastructure has been upgraded and leaks have been significantly
reduced. The additional site visits in 2023 also found that several open-ended lines detected in 2019 had indeed been sealed.

The four scenarios employed to modify the emission distribution reflect different possible effects of emission mitigation:
The most pessimistic is Scenario 1 which assumes that only the emissions from the largest sources were curbed while those
below the detection limit of AVIRIS-NG remained unaffected. The most optimistic is Scenario 4 which assumes that all
emissions were reduced equally. This is quite unlikely given the considerable effort of controlling all production sites rather
than addressing only the largest leaks. Unfortunately, direct emission measurements below the AVIRIS-NG detection limits are
lacking in 2021 to constrain these scenarios, but they encompass the full range of potential mitigation efforts and are supported
by site visits in subsequent years.

The situation at the processing facilities is complex. The number of super-emitters observed in 2021 is lower than expected
from the measurements conducted in 2019. However, the uncertainty in this estimate is large due to the limited sample size. We
have strong evidence that at least one super-emitter (C2) was closed after the AVIRIS-NG flights and it is possible that others
were addressed already before then. A complication is the possibility that mitigation measures at the production sites (e.g.,
sealing of open-ended lines) might increase the emissions from venting in the surroundings, which according to our study, was
the largest source of emissions at and near processing facilities. In 2021, we identified six "open field" sources that we assigned
to nearby processing facilities. Four of these were identified during the 2022 survey as vent stacks, which were already visible
in Google imagery ten years earlier (A1, A3, C2, and B10). Due to the lack of information regarding the location and number
of these stacks in the dataset provided by the operator, these sources were not visited during the ground-based campaign in
2019. Since venting is generally associated with high emissions, the emission factors estimated from the 2019 measurements
might be biased low. Additional measurement campaigns would be necessary to better constrain emissions in Romania. While
we see some evidence for mitigation at processing facilities, the estimated emissions remain highly uncertain.

Our top-down estimates in the study area still exceed emissions reported to the UNFCCC for the entirety of Romania. In
particular, we find that reported emissions from oil production (Fig. 7b) are substantially lower than our measurement-based
estimates. In contrast, reported emissions from gas production are higher than top-down estimates, likely because most gas
production occurs outside our study area in northern Romania. In the reported emission, $CH_4$ emissions from venting and
flaring are mainly due to oil production. These emissions can occur at oil production sites or at processing facilities. If all
venting and flaring would occur at production sites, the reported emissions would be consistent with our measurement-based
estimates (Scenarios 3 and 4). However, while the ground-based survey in 2019 found venting primarily at production sites,
the small number of super-emitters at production sites and the high emissions from vent stacks in 2021 indicates that venting
shifted towards processing facilities. In this case, top-down estimates at oil production sites would be substantially greater than
reported emissions, suggesting that the emission factors used to compile the reports are too low. However, this assumes that

the diurnal variability of emissions is neglectable, while studies in other regions show evidence for a diurnal cycle in emissions (e.g., Vaughn et al., 2018). Our measurements were conducted exclusively during daytime, so additional measurements would be necessary to constrain the diurnal cycle of emissions.

## 5 Conclusions

Our study highlights the importance of monitoring emissions using measurement-based techniques to improve emission estimates, identify their main causes, and monitor the effectiveness of mitigation measures. It also demonstrates the need to
combine large-scale mapping surveys with instruments like AVIRIS-NG with ground-based surveys to cover the full range of sources from small leaks to super-emitters. Despite the high detection limit, AVIRIS-NG directly measured *in only two days* $CH_4$ emissions of 2964 kg/h (i.e. 26 kt), which is 15-34% of our bottom-up estimates for the basin and 40% of the reported emissions for the *entirely* of Romania. The emissions originated at only 25 locations, thus providing targets for effective and efficient future reduction of emissions in Romania. Additional ground-based measurements and airborne campaigns, e.g. with the
new AVIRIS-4 instruments with better detection limits (Green et al., 2022), would be needed to ensure continuous monitoring of emission reduction measures in the future.

*Code and data availability.* The codes for $CH_4$ retrieval and upscaling are available on request. The ddeq Python library for data-driven emission quantification (Kuhlmann et al., 2024) is available on Gitlab.com (https://gitlab.com/empa503/remote-sensing/ddeq). AVIRIS-NG Level 1 data are available on the AVIRIS-NG data portal (https://avirisng.jpl.nasa.gov/dataportal/). The data used for upscaling are available
in the supplement. The AVIRIS-NG $CH_4$ maps and the results from the emission quantification are published on the Zenodo data repository (DOI: 10.5281/zenodo.14054126)

*Author contributions.* GK developed and applied the methane retrieval and the emission quantification methods. GK, FS and DZA developed the methods for merging and upscaling the 2019 and 2021 datasets. SS, AT and AH collected and prepared the raw data used in the study. The study was conceptualized by GK, AC, TR and DB and supervised by TR and DB. Funding was acquired by SS, LE, AC, TR and DB.
The paper was prepared and revised by GK with input from all co-authors.

*Competing interests.* The authors declare no competing interests.

*Acknowledgements.* We acknowledge the CHIME-SBG Hypersense campaign by the European Space Agency for providing the framework in which to operate AVIRIS-NG in Europe during the summer of 2021. We would like to thank "Intero - The Sniffers" for their follow-up visits to the sites.



*Financial support.* This research has been funded under the framework of UNEP's International Methane Emissions Observatory (IMEO). The ROMEO campaign in 2019 was initiated and largely carried out by participants of the European H2020 project MEMO2, which was funded by the European Union's Horizon 2020 research and innovation program under the Marie Skłodowska-Curie grant agreement no. 722479. AVIRIS-NG flights were supported by the Jet Propulsion Laboratory and NASA's Earth Science Division from funding provided by Carbon Mapper to augment planned flights in Europe.



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
