# Peer review of "Evidence of successful methane mitigation in one of Europe's most important oil production region"

_EGUsphere, 2024_

## Author Comment (AC1)

The manuscript is an interesting showcase of the impact of a measurement campaign. It is important to highlight the combination of so many techniques and measurement methodologies. The authors have been capable to include the point source measurements, distributions of emissions and global emissions budgets. They also compared to inventories and explored different hypothesis for emission reductions. Although this is the most remarkable part of the manuscript, it is also the most challenging part since many areas need to be effectively covered and reviewed. Thus, in some cases, the manuscript needs some clarifications/review.

**Reply:** We like to thank Reviewer #1 for their positive and constructive comments. In the following, we address the major and minor comments point by point.

The major/minor points that are suggested for improvement are the following:

1. Page 2 line 29. Clarify here (e.g. with a footnote) what you define as a Some might set the threshold at 10kg/h, other at 100kg/h. The latter typically for satellite studies.

   **Reply:** We would define super-emitters not in terms of emission thresholds, but rather as those sources in the high-emitting tail of the emission distribution that contribute a major proportion of total emissions, i.e. 10 kg/h for oil and gas in Romania. We have modified the sentences as follows:

   "it is therefore critical to detect and quantify a statistically robust number of ``super-emitters'', i.e. those sources in the high-emitting tail of the emission distribution that contribute significantly to the total emissions. For the O&G sector in Romania, this threshold is about 10 kg/h. Identifying a robust number of super-emitters is challenging with ground-based surveys ..."

2. Page 2 line 42. Do you refer to Fig1 entirely? in that case, it should be Fig1 rather than Fig1a.
   **Reply**: We changed this to "Figure 1".

3. Page 2 line 44. Do you mean it has same spectral sampling/resolution? you can clarify writing down "5nm spectral sampling and resolution"

   **Reply**: Changed as follows: "The spectrometer samples a spectral range from 380 to 2510 nm at 5 nm with a spectral resolution of 5 to 7.5 nm full width at half maximum (FWHM). The spectral range includes two spectral windows...

4. Page 4 line 84. Please specify the dataset/web where you got the values of CO2 and CH4

   **Reply**: $CH_4$ is the globally averaged concentration reported by NOAA Global Monitoring Laboratory for 2021 (https://www.noaa.gov/news-release/increase-in-atmospheric-methane-set-another-record-during-2021). We used the same source for the $CO_2$ concentration, which varied between 414 and 416 ppm in 2021. The reference was added to the manuscript.

5. Equation (7) it can be assumed that the equation is used to generate Lo and Leps; consequently retrieving s(lambda). Please, clarify in the text how you effectively use the equation.

   **Reply**: Yes, Eq. (7) was used to compute the at-sensor radiances L0 and Leps. We have modified the text to clarify how we compute the unit absorption spectrum.

6. Page 4 line 93 The authors are correct since more bands implies a more demanding spectrum to match but can also lead to an underestimation of the methane signal. Can you further discuss this point and how it affects the AVIRIS measurements? you have a good discussion here https://doi.org/10.5194/amt-17-1333-2024

   **Reply**: We are aware that a wide windows as used by Roger et al. (2024) can result in underestimation of the $CH_4$ enhancement. We therefore used two spectral windows, i.e. 1480-1800 nm and 2080-2450 nm, simultaneously, instead one wide window. We tested this with synthetic spectra and did not find evidence for an underestimation using this approach. We have clarified this in the manuscript and added the reference.

7. Page 6 line 120 please specify that this enhancement must be transformed from concentrations ppm *m to g/cm2 or similar units.

   **Reply**: We now specify that we convert enhancements to $kg/m^2$.

8. Page 6 line 122 From Kuhlmann 2024 the effective wind speed is taken from the provided wind speeds at the source location. It seems you directly take the wind speed U10 as Ueff and consider the height as an uncertainty in the budget. Do you take a spatial or temporal interpolation? just the single pixel?

   **Reply**: We interpolate the hourly analysis product from 10 km to the source location using the nearest neighbor method. We did not find a significant change of the results using linear interpolation.

9. Page 6 line 122 From Kuhlmann 2024 a decay time can be provided to compute the decay time correction term. Why is not applied? are there significant differences?

   **Reply**: The correction term is only necessary for gases with lifetimes relevant at plume scale (minutes to hours), which is not the case for CH4 with a lifetime of several years.

10. Page 8 line 180. Whereas the separation into systematic and random components is positive, it might be more adequate to distinguish between spatially correlated and uncorrelated components. Thus, we can connect the impact of CH4 uncertainty in the final flux rates.

    **Reply**: We have revised the section and now use 'error' instead of 'systematic error' or 'random error' and describe whether the errors are spatially and temporally correlated and the effect this has on the final flux rates.

11. Figure 3 is a very interesting exercise. It shows important errors for parameters such as SZA, VZA or AOD. These three parameters are directly linked to the previous questions on the model (equation 7). However, these are systematic known errors and theoretically should be directly compensated. Since this is not the case, you can estimate the specific error per plume and add them linearly (not quadratically) in the expanded uncertainty budget (see JCGM 100:2008 GUM 1995 with minor corrections). Please, specify your methodology to include them in the uncertainty budget.

    **Reply**: The exercise shows that using the mean value for SZA, VZA and surface elevation for a line result in a quite small error (<5%). However, it is true that this error can be spatially correlated inside a plume. We account for this now by increasing the uncertainty of the integrated mass by 5%.

We also added a note that the SZA shown in Fig. 3b is the range for the full campaign, but SZA does not vary significantly for an AVIRIS-NG line.

12. Page 8 line 186.Once the pixels are aggregated into an integrated mass, we would expect that the systematic component dominates whereas the random one is highly reduced due to spatial error correlation. Please, clarify why you consider the random error component of CH4 map.

    **Reply**: We now clarify that we assume spatially correlated errors in CH4, which add up when computing the integrated mass. We thus increase the uncertainty of the integrated mass by 5%.

13. Page 8 line 185. the plume length includes an uncertainty itself as correctly explained here but this uncertainty needs to be propagated to Q. Please specify this and how the correlation between IME and L might partially compensate.

    **Reply**: To estimate the uncertainty of Q, we propagate uncertainty for IME, L and U assuming uncorrelated errors. We added the equation used for computing the uncertainty in the manuscript. We did not account for the correlation between IME and L, but since we do not compute L from the detectable area and IME includes some background values, the correlation is less strong as for other implementation of the IME method.

14. Page 9 line 197. the ERA5 ensemble only provides a small range of sensitivities. https://confluence.ecmwf.int/display/CKB/ERA5%3A+uncertainty+estimationunfortunately, most systematic (and dominant) effects are not taken into account. It is very positive that height dependency is considered but important effects such as the U10 spatiotemporal representativeness and ERA5 modelling errors are not included. Please review this uncertainty contribution and clarify the final figures.

    **Reply**: The ERA5 ensemble spread results in an uncertainty estimate of about 0.3 m/s. It is true that this only provides a limited estimate of the model error and does not include the uncertainty due to interpolation from the model fields (10 km hourly) to the source location. To account for this, we now estimate the wind speed uncertainty from a comparison of the model fields with wind observations in the study area. The uncertainty was estimated to be 1.0 m/s, which is consistent with evaluation studies of ERA5 winds with observations (e.g., Potisomporn et al. 2023; Vanella et al. 2022) that estimate RMSDs ranging from 1-2 m/s. As we did not find a significant bias between model and observations, we assume that the uncertainties from wind speed are not correlated between sources.

15. Page 9 line 223. It should be better clarified which assumptions are used to combine the uncertainty components in the MonteCarlo simulation. Specifically, the combination of the different AVIRIS uncertainty estimates.

    **Reply**: We have restructured Section 2.6 merging Sections 2.6.1, 2.6.5 and 2.6.6 to better describe the steps for computing emission estimates. We also added a new section 2.6.5 that describes the estimation of uncertainty using the Monte Carlo simulations. We added that the uncertainties of the AVIRIS-NG estimates are added assuming uncorrelated errors, because the uncertainty is dominated by the uncertainty in the wind speed, for which no bias was found.

16. Page 12 line 286. you measured the same spot three times and, it is assumed, in relative short time. Did you find consistency? it would be very positive to disclose this information for a more robust validation and understanding of the emission source potential changes.

    **Reply**: Yes, the individual estimates are consistent within the estimated uncertainties. We have updated Figure 5 to also show the individual estimates for each source. The individual estimates are already available in the supplement on the figures for each source. The raw data have also been published (https://doi.org/10.5281/zenodo.12773375). The dataset will be updated to account for the changes in the computation of the uncertainty.

17. Figure 6d this image shows an outlier following the wind direction. Is the continuation of the detected plume? Is there any explanation for it?

    **Reply**: This is an interesting case. We originally assumed a second source at the location of the second maximum. However, this second source was not detectable on the flight on the previous day (see Fig. S17). There is also no evidence for any vent stack on Streetview at the location. We therefore think that this a continuation of the detected plume with a small gap due to turbulence or inconsistent emissions at source.

18. Page 16 line 365 Are there any difference in wind speeds between 2019 and 2021? although unlikely explaining all these differences, it could be helpful to support the discussion.

    **Reply**: We are not sure we understand this point. The campaign in 2019 employed in situ instruments over three weeks, which much higher detection limit than AVIRIS-NG.

---

## Author Comment (AC2)

The manuscript presents airborne remote sensing observations and quantification of methane plumes in Romania over two days in 2021. The findings are compared with previous ground-based and drone measurements from 2019, and a total emission estimate for the region is calculated. The study discusses the advantages and limitations of each measurement technique, concluding that methane emissions in the region have decreased between the two studies. While comparing the results is challenging due to the variability in the oil and gas production landscape and differences in the design of the studies, the authors have made a commendable effort to address these complexities.

*Reply*: We like to thank Reviewer #2 for their positive and constructive comments. In the following, we address the comments point by point.

We have the following comments that need clarification/correction:

1. Line 32: How do you define super-emitters? Are all AVIRES-NG detections declared super-emitters here?

   **Reply**: We define super-emitters as those sources in the high-emitting tail of the emission distribution that contribute a major proportion of total emissions, i.e. 10 kg/h for oil and gas in Romania. See also our reply to comment 1 by Reviewer #1.

2. Line 122: The two profiles here are theoretical and not used in this study. Please clarify here.

   **Reply**: We have rewritten the paragraph to better clarify how the effective wind speed is computed in our study. We also moved the description of the wind uncertainty completely to the relevant section.

3. Line 123: change "a" to "at".

   **Reply**: Done.

4. Line 124: The local wind speed uncertainty might be very different from an uncertainty at 10 km resolution. Please comment.

   **Reply:** We now include an estimate of the uncertainty due to interpolation from the model fields (10 km hourly) to the source location of 1.0 m/s. This value was estimated by comparing the model fields with wind observations in the study area. See also our reply to comment 14 by Reviewer #1.

5. Line 185: Where does this assumption stem from?

   **Reply**: We now clarify how this was estimated: "The uncertainty of the plume length was set to 10% and to at least half a pixel size (i.e. about 5 m), which is a rough estimate considering how the plume length can vary when modifying the threshold for the plume detection algorithm."

6. Line 186: "plumes"

   **Reply**: Done.

7. Line 197: Where do the 15% for the height-dependency uncertainty stem from?

   **Reply** We estimate this number by computing the wind speed at 5 m and 20 m for the logarithmic wind profile, which is 15% lower and higher than the wind speed at 10 m. This is also described in the text.

8. Line 200: "would be"

**Reply**: Done.

9. Line 200: What is the contribution of wind speed uncertainty to the total uncertainty?

   **Reply:** After revising the estimate of uncertainty for wind speed, the average contribution of wind speed to the total uncertainty is about 80%. We added the following to the manuscript:

   "The mean uncertainty of the estimated emission rate is 84% ranging from 40 to 241%. Our total uncertainty in the estimated emissions is dominated by the uncertainty in the wind speed, which has a mean uncertainty of 76% (32-239%). The mean uncertainty in the integrated mass, i.e. the uncertainty in the CH4 retrieval, is 25% (9-64%)."

10. Line 249: remove "that"

    **Reply**: Done.

11. Line 252: Please explain why you chose these scenarios and what they infer for the change in emissions.

    **Reply**: We have added an explanation of the scenarios to the method section: "Scenario 1 uses the mean distribution from 2019 and Scenario 2 uses the distribution from 2019 that corresponds to the lower limit of estimated emissions. Scenario 1 and 2 assume that the emission distribution is still valid below the AVIRIS-NG detection limit, meaning any emission reduction results from fewer than expected AVIRIS-NG detections. Scenario 3 and 4 assume that emissions also changed below the AVIRIS-NG detection limit. To account for this, we reduce the standard width σ for Scenario 3 and mean μ for Scenario 4 of the 2019 distribution such that the number of expected emitters above the detection limit matches the six emitters found with AVIRIS-NG. Scenario 3 reduces the width of the emission distribution meaning there are fewer low and high emitting sources. Scenario 4 shifts whole emission distribution assuming all emissions were reduced equally, instead of addressing only the largest sources."

12. Line 264: The table states and EF of 1.2 for scenario 4.

    **Reply**: Thank you for catching this. We changed to the correct value of 1.2.

13. Line 293: That is quite a high contribution from "unassigned" sources. Are these findings considered when relating to 2019 findings?

    **Reply**: 6 out of 12 sources in the open field were classified as "unassigned" sources. The other 6 sources were assigned to facilities in the proximity. The unassigned sources are considered as own category contributing 9 kt of the total emissions.

14. Figure 5: Please show the values of the individual emissions estimates that contribute to the mean emissions.

    **Reply**: We have added the individual estimates to the figure.

15. Line 364: "a robust estimate"

    **Reply**: Done.

16. Line 410: Replace "surveys with instruments" with "surveys using instruments"

    **Reply**: Done.

17. Line 413: Change "entirely" to "entirety"

    **Reply**: Done.